# Prospective Association of the Portfolio Diet with All-Cause and Cause-Specific Mortality Risk in the Mr. OS and Ms. OS Study

**DOI:** 10.3390/nu13124360

**Published:** 2021-12-03

**Authors:** Kenneth Lo, Andrea J. Glenn, Suey Yeung, Cyril W. C. Kendall, John L. Sievenpiper, David J. A. Jenkins, Jean Woo

**Affiliations:** 1Department of Applied Biology and Chemical Technology, The Hong Kong Polytechnic University, 11 Yuk Choi Road, Hung Hom, Kowloon, Hong Kong SAR, China; 2Research Institute for Smart Ageing, The Hong Kong Polytechnic University, Hong Kong SAR, China; 3Department of Nutritional Sciences, Temerty Faculty of Medicine, University of Toronto, Toronto, ON M5S 1A8, Canada; andrea.glenn@utoronto.ca (A.J.G.); cyril.kendall@utoronto.ca (C.W.C.K.); john.sievenpiper@utoronto.ca (J.L.S.); david.jenkins@utoronto.ca (D.J.A.J.); 4Clinical Nutrition and Risk Factor Modification Center, St. Michael’s Hospital, Toronto, ON M5C 2T2, Canada; 5Toronto 3D Knowledge Synthesis and Clinical Trials Unit, St. Michael’s Hospital, Toronto, ON M5C 2T2, Canada; 6Department of Medicine and Therapeutics, The Chinese University of Hong Kong, Shatin, New Territories, Hong Kong SAR, China; sueyyeung@cuhk.edu.hk; 7College of Pharmacy and Nutrition, University of Saskatchewan, Saskatoon, SK S7N 5E5, Canada; 8Department of Medicine, Temerty Faculty of Medicine, University of Toronto, Toronto, ON M5S 1A8, Canada; 9Li Ka Shing Knowledge Institute, St. Michael’s Hospital, Toronto, ON M5B 1A6, Canada; 10Department of Medicine, Division of Endocrinology and Metabolism, St. Michael’s Hospital, Toronto, ON M5C 2T2, Canada

**Keywords:** Portfolio Diet, Asian population, mortality, cardiovascular disease, cancer, prospective cohort

## Abstract

The Portfolio Diet has demonstrated its cardiovascular benefit from interventions, but the association between Portfolio Diet adherence and the risk of all-cause and cause-specific mortality has not been examined in Chinese population. The present study has collected Portfolio Diet adherence (assessed by food frequency questionnaire), lifestyle factors and mortality status of 3991 participants in the Mr. Osteoporosis (OS) and Ms. OS Study. Cox regression models were used to examine the association between the Portfolio Diet adherence and mortality risk (all-cause, cardiovascular disease or cancer). The highest quartile of the Portfolio Diet score was associated with a 28% lower risk of all-cause (hazard ratio, HR: 0.72) and cancer (HR: 0.72) mortality, respectively. The association between Portfolio Diet adherence and cardiovascular disease mortality did not reach statistical significance (HR: 0.90, 95% CI = 0.64, 1.26). Among male participants, the highest adherence to the Portfolio Diet was also associated with a lower risk of all-cause (HR: 0.63) and cancer mortality (HR: 0.59), and there was an inverse association between food sources of plant protein and the risk of cardiovascular mortality (HR: 0.50). However, most associations between the Portfolio Diet and mortality were not significant among females. The protection for cancer mortality risk might reach the plateau at the highest adherence to the Portfolio Diet for females. To conclude, greater adherence to the Portfolio Diet was significantly associated with a lower risk of mortality in Hong Kong older adults, and the associations appeared stronger among males.

## 1. Introduction

Maintaining a healthy diet is a major lifestyle factor to prevent diabetes and cardiovascular diseases (CVD), two chronic diseases that have resulted in a huge public health burden worldwide. Although multiple studies have evaluated the relationship between individual nutrients and health, emerging evidence has suggested how dietary patterns may explain the interactions between different foods and food components and their associations with health outcomes [1]. Some well-known dietary patterns include the Mediterranean and the Dietary Approaches to Stop Hypertension (DASH) diets [2,3].

Amongst dietary patterns that may improve cardiovascular health, the Dietary Portfolio, or the Portfolio Diet, is a plant-based dietary pattern developed in the early 2000s to lower low-density lipoprotein cholesterol (LDL-C) [4,5]. The Portfolio Diet is low in saturated fat and cholesterol, as in the National Cholesterol Education Program Adult Treatment Panel III (NCEP ATP III) diet [6], with added plant-based foods or nutrients with cholesterol-lowering properties, namely nuts, plant protein (soy and pulses), viscous fiber sources (oats, barley, psyllium, eggplant, okra, and recently apples, oranges and berries), plant sterols and monounsaturated fats (olive oil, sunflower oil, canola oil, avocado). In 2018, a meta-analysis of randomized controlled trials showed that the Portfolio Diet significantly lowered LDL-C by 17% more than the NCEP ATP III diet alone [7].

In addition to clinical trials for patients with dyslipidemia, the health benefits of the Portfolio Diet were also examined among cohort studies. A Portfolio Diet score (PDS) has been developed and validated to assess the adherence to the Portfolio Diet using food frequency questionnaires (FFQ) [8], and to date, the PDS has been associated with lower cardiometabolic risk factors in the Prevención con Dieta Mediterránea (PREDIMED)-Plus cohort and lower risk of CVD in the Women’s Health Initiative (WHI) [9,10].

Despite groundwork being established, the PDS needs to be assessed in other cohorts and ethnic groups to compare with previous findings and verify its applicability internationally. In addition, the previous work using WHI data only included postmenopausal women. The present study has evaluated how the adherence to the Portfolio Diet may have sex-specific associations with the risk of all-cause, CVD and cancer mortality among a prospective cohort of Hong Kong older adults, an ageing Chinese population with high life expectancy [11]. When analyzing the secular trends of cancer incidences in 1983–2017, the total incident cancer case counts increased 156.5% for women and 96% for men, which might be due to population growth (66.1% for women, 25.4% for men) and population ageing (95% for women, 119.4% for men) [11]. That would be important to evaluate how a healthy dietary pattern may reduce the mortality risk due to chronic diseases for older adults.

## 2. Materials and Methods

### 2.1. Study Population

To measure the Portfolio Diet adherence in the Hong Kong Chinese context, we have used data from the Mr. Osteoporosis and Ms. Osteoporosis Study (OS Study, hereafter). Since 2001–2003, the OS Study has been a long-term cohort study that recruited 4000 community-living Hong Kong Chinese men and women aged at least 65 years. The target was to recruit a stratified sample that equally assigned them into each of these age groups, namely 65–69, 70–74, and ≥75. Compared with the general population in this age group, participants had higher educational level (9.8% vs. 3.8% with tertiary education), higher proportion of being married (70.7% vs. 59.9%) and slightly lower proportion of living alone (10.8% vs. 11.3%) [12]. The study was performed based on the guidelines set in the Declaration of Helsinki and was approved by the Clinical Research Ethics Committee of The Chinese University of Hong Kong. Written informed consent was obtained from all participants. Dietary assessment has been administered as an FFQ to all participants at baseline, and the data on mortality statistics has been obtained from the Death Registry collected through March 2014 [12].

### 2.2. Dietary Assessment

Dietary assessment was conducted using a validated FFQ (OS FFQ, hereafter) with 280 food items validated against 24 dietary recalls [13,14]. Each participant was asked to respond to the FFQ by trained personnel by answering how often they consumed each food item each day or each week, and the size of each portion, over the past year. A pictorial catalogue with individual food portions was provided as a guide.

Food items recommended in the Portfolio Diet were extracted from the OS FFQ and were categorized into the six components of the Portfolio Diet. For most components, servings per day were summed over all consumed food items in each component for every participant. More points were given to participants with higher intakes of food recommended in the Portfolio Diet, whereas fewer points were given to participants with higher intakes of food not recommended in the Portfolio Diet. These points were given for each of the six dietary components (plant protein sources, nuts, viscous fiber, plant sterols, monounsaturated fatty acids (MUFA), saturated fat/cholesterol sources) by splitting the components into quintiles (according to the sex-specific data distribution in the OS Study). Those in the highest quintile of foods recommended (e.g., nuts) received 5 points and those in the lowest quintile received 1 point. Reverse scoring was done for those with foods not recommended (e.g., foods high in saturated fat), meaning those with the highest intake (quintile 5) received 1 point, and those with the lowest intake (quintile 1) received 5 points. The total points were then added for each respondent, resulting in a score range between 6 and 30, with higher scores indicating higher adherence to the Portfolio Diet. The adherence score of Portfolio Diet has been validated in 652 participants in the Toronto Healthy Diet Study, which has shown inverse association with lower LDL-C, and showed reasonable agreement between the score from the FFQ and 7-day dietary record [8].

Plant sterols were the only score component based on mg per day, while all other components are food-based in servings per day. Since the OS FFQ compositional database does not have plant sterols available as a nutrient variable for their FFQ, we have established a plant sterol database (mg per day) based on literature values. Data sources in European countries or the United States were used, including the Finnish Food Composition Database [15], the database used in the European Prospective Cohort into Cancer cohort [16], and the United States Department of Agriculture [17]. For food items specific to the Asian context, we obtained the nutritional values from food composition databases or academic publications [18,19,20]. We created recipes and determined plant sterol values for foods in the OS FFQ that does not have a plant sterol value available in the literature, using the ESHA Research Food Processor SQL: Nutrition Analysis and Fitness Program (Copyright 2012, ESHA Research).

### 2.3. Ascertainment of Mortality Outcomes

Data on mortality were obtained from the Death Registry of the Department of Health of Hong Kong and collected through March 2014 with a median follow-up of 11.1 years. Apart from all-cause mortality, cardiovascular or cancer causes of death were identified by the cause of death reported on the death certificate and classified according to the International Classification of Disease version 10 codes. Participants were followed up until death or loss to follow-up, whichever occurred first [13].

### 2.4. Covariates

Demographic, lifestyle and health information was collected from each participant including education level (secondary school or below vs. post-secondary education), smoking habit (never, former or current smoker), alcohol consumption (drink > 12 alcohol drinks in the previous year), physical activity, dietary intake and self-reported history of chronic diseases (diabetes, hypertension, stroke, heart attack, angina, congestive heart failure or cancer). Physical activity level was evaluated using the Physical Activity Scale for the Elderly (PASE) [21]. Body weight was measured using the Physician Balance Beam Scale (Healthometer, IL, USA) to the nearest 0.1 kg with participants wearing a light gown. Height was measured to the nearest 0.1 cm using the Holtain Harpenden Standiometer (Holtain Ltd., Crosswell, UK), which were used to compute body mass index (BMI).

### 2.5. Statistical Analysis

Baseline characteristics were stratified by sex (male or female) and the quartiles of the PDS using means with standard deviations for continuous variables and frequencies with percentages for categorical variables. To compare baseline characteristics, chi-square tests were used for categorical variables and analysis of variance for continuous variables. Participants were categorized into quartiles of the Portfolio Diet score, with the lowest quartile serving as the reference group. Cox proportional hazard models were used to estimate hazard ratios (HRs) and 95% confidence intervals (CIs) for the association between the PDS quartiles and mortality outcomes. The Cox regression model was adjusted for sex (male vs. female), age, dietary energy, body mass index, physical activity, systolic blood pressure (all continuous variables), medical history (yes vs. no for diabetes, hypertension, stroke, heart attack, angina, congestive heart failure or cancer), smoking habit (never, former or current smokers), alcohol drinking (yes vs. no for having >12 alcoholic drinks in the past year) and education level (post-secondary education vs. secondary school or below). We also applied subgroup analyses according to sex (male or female). Trend analysis was performed by assigning median values of each PDS quartile and treating it as a continuous variable in the regression model. To examine the potential non-linear relationship between the PDS and the risk of mortality, we compared the fit of continuous models with or without cubic spline terms of PDS. A likelihood test with *p* < 0.05 would suggest a better fit regression model by including the quadratic term, hence a non-linear relationship between the PDS and all-cause, CVD or cancer mortality.

Additional analyses included evaluating associations between the six individual components of the Portfolio Diet and risk of the mortality outcomes. We conducted subgroup analysis on the Portfolio Diet components using the same covariates as listed in the regression analysis. Sensitivity analyses on the association between the PDS and the risk of mortality were conducted by excluding people who had ever smoked or with individual comorbidity at baseline (diabetes, hypertension, stroke, heart attack, angina, congestive heart failure, cancer, or any one of the above) as they may have different diets than those without smoking nor comorbidities. Since the influence of the PDS on blood pressure could be the causal pathway on the risk of mortality, we also removed systolic blood pressure from the regression model in a sensitivity analysis. Another sensitivity analysis comprised using the age at death as the survival time in the Cox regression model, instead of the observation period from enrolment to the end of the study or the subject’s death, because the exposure (Portfolio Diet adherence) for each subject began before the period of enrolment. With regard to the competing nature of cardiovascular disease and cancer in mortality risk, we have also computed the cumulative sub-distribution hazards (SHR) for the Fine–Gray models [22], and compared the magnitude of association with that obtained from the standard Cox models. The sex-specific differences in cumulative incidence of mortality due to CVD or cancer were compared using the approaches proposed by Gray [23]. Statistical tests were two-sided and *p* < 0.05 was considered statistically significant. Most statistical analyses were conducted with IBM SPSS Statistics 23 (IBM Corporation, Armonk, NY) and we have used R 4.1.1 (R Foundation for Statistical Computing, Vienna, Austria) for examining the non-linear relationship between the PDS and the risk of mortality, and the competing risk analysis [22].

## 3. Results

### 3.1. Characteristics of OS Study Participants

After excluding 9 people with missing information on dietary intake or covariates, a total of 3991 participants were included in the analyses. The baseline characteristics of participants in the OS Study are presented by sex in Table 1. As shown by independent t-test, male participants tended to have, on average, higher dietary energy, lower BMI, higher physical activity and lower SBP compared to female participants. Male participants also had higher consumption in food sources rich in plant protein, nuts, plant sterols and saturated fat/cholesterol, but lower consumption in food sources rich in viscous fiber and MUFAs. As demonstrated by the chi-square test, males were more prevalent in stroke, angina, being a smoker or alcohol drinker, and having post-secondary education compared to females. Appendix A provides the details on the mean intake of each Portfolio Diet components by sex-specific quintiles.

The baseline characteristics of participants in the OS Study by PDS quartiles are presented in Table 2. As shown by one-way ANOVA, participants in the highest quartile of PDS were, on average, more likely to have post-secondary education, be younger in age, have a higher physical activity level and higher intake of energy, and lower smoking rate. but the rate of co-morbidities was similar across PDS quartiles. We have also compared the sex-specific cumulative incidence of mortality due to CVD or cancer in Appendix A, and did not find significant differences in incidence by sex (both *p* > 0.05).

### 3.2. Portfolio Diet Score and Mortality Outcomes

The association between the Portfolio Diet adherence in quartiles and the risk of mortality are presented in Table 3. When compared to participants with the lowest adherence, the highest adherence to the Portfolio Diet was associated with a lower risk of all-cause mortality (HR: 0.72, 95% CI = 0.62, 0.86) and cancer mortality (HR: 0.72, 95% CI = 0.54, 0.96). The association between the PDS and CVD mortality did not reach statistical significance (HR: 0.90, 95% CI = 0.64, 1.26). Among male participants (Table 3), the highest adherence to the Portfolio Diet was also associated with a lower risk of all-cause mortality (HR: 0.63, 95% CI = 0.51, 0.79) and cancer mortality (HR: 0.59, 95% CI = 0.39, 0.87). Among females, participants in the second (HR: 0.65, 95% CI = 0.43, 0.98) and third (HR: 0.62, 95% CI = 0.40, 0.95) quartiles of PDS had associations with a lower risk of cancer mortality. The *p*-values for the trend were significant for the association between PDS and all-cause mortality among all participants and males only, as well as the association between PDS and cancer mortality among overall participants. The *p*-values for non-linearity were significant for the association between PDS and all-cause mortality among males, and the association between PDS and cancer mortality among males and females. We then checked the assumption by testing the proportionality of the risks according to outcomes (all-cause, CVD or cancer mortality) and sex (male, female). As revealed by the Schoenfeld test in Appendix A, most Cox regression models did not violate the assumption except for the association between PDS and all-cause mortality among all participants (*p* = 0.0199).

When looking into individual components of the Portfolio Diet (Table 4), higher consumption of plant protein sources (HR: 0.77, 95% CI = 0.62, 0.95), viscous fiber sources (HR: 0.78, 95% CI = 0.62, 0.98) and nuts (HR: 0.73, 95% CI = 0.58, 0.91) were associated with a lower risk of mortality among males. There was also an inverse association between food sources of plant protein and the risk of cardiovascular mortality (HR: 0.50, 95% CI = 0.30, 0.81) among male participants. However, associations between adherence to the Portfolio Diet score and each mortality outcome were not significant among females.

### 3.3. Sensitivity Analysis and Competing Risk Analysis

After excluding participants who smoked or with comorbidities at baseline, excluding SBP from the regression model, or using the age at death as survival time, the association between the highest quartile of PDS and the risk of mortality remained significant among male participants (Appendix A). Among female participants, the association between the highest quartile of PDS and the risk of mortality remained insignificant in most sensitivity analyses, except the significant association between PDS, all-cause (HR: 0.62, 95% CI = 0.42, 0.92) and cancer mortality (HR: 0.52, 95% CI = 0.27, 0.98) after excluding participants with hypertension (Appendix A). The association between PDS and CVD mortality was insignificant in all sensitivity analyses regardless of sex (Appendix A). As demonstrated from competing risk analysis on the association between PDS and the risk of mortality due to CVD and cancer (Appendix A), the association between the PDS and CVD mortality did not reach statistical significance regardless of sex. Among male participants, the third (SHR: 0.88, 95% CI = 0.52, 0.93) and the highest (SHR: 0.54, 95% CI = 0.43, 0.67) adherence to the Portfolio Diet associated with a lower risk of cancer mortality. Among females, participants in the second (SHR: 0.64, 95% CI = 0.42, 0.97) and third (SHR: 0.60, 95% CI = 0.39, 0.93) quartiles of PDS had associations with a lower risk of cancer mortality. The magnitude of association did not have a substantial difference when compared to the results obtained from standard Cox regression model (Table 3).

## 4. Discussion

### 4.1. Summary of Findings

This examination of approximately 4000 older adults in Hong Kong is the first prospective cohort study to examine the association between the Portfolio Diet and mortality in the Asian region. A higher Portfolio Diet score was associated with a 28% lower risk of all-cause and cancer mortality, respectively. Among male participants, the highest adherence to the Portfolio Diet was also associated with a lower risk of all-cause and cancer mortality, and there was an inverse association between food sources of plant protein and the risk of cardiovascular mortality. Since most associations were not significant among female participants, the findings from the present study are likely driven by male participants. Sex differences in the consumption of food sources rich in plant protein, viscous fiber and nuts, as well as smoking rate, may explain the relationship between the Portfolio Diet and mortality risk reduction. Despite some non-linearity in the shape of the relationships, the general results from the present study have suggested protective associations of the Portfolio Diet in decreasing the risk of mortality. The protection for all-cause and cancer mortality risk may be significant at the highest quartile of PDS for males, while the protection for cancer mortality risk might reach the plateau at the highest quartile of PDS for females.

### 4.2. Comparison with Previous Literature

To the best of our knowledge, this study is the first to demonstrate the potential role of the Portfolio Diet in preventing all-cause mortality; therefore, comparison with other literature is challenging. An analysis of postmenopausal women that participated in the WHI showed that the adherence to the Portfolio Diet was associated with lower risk of total CVD (HR: 0.89, 95% CI: 0.83, 0.94), coronary heart disease (HR: 0.86, 95% CI: 0.78, 0.95), and heart failure (HR: 0.83, 95% CI: 0.71, 0.99), comparing the highest to lowest quartile of adherence [10], which is not in line with the findings from our study. In other words, the relatively small sample size (3991 older adults when compared with 123,330 postmenopausal women in WHI) for the present study might limit the statistical power to detect associations between PDS and CVD mortality. Future studies should verify the long-term association between PDS and CVD outcomes in larger prospective cohorts.

Although the present study has not demonstrated the long-term benefit of the Portfolio Diet in reducing mortality risk due to CVD, further recent analyses revealed the prospective benefit of the adherence to the Portfolio Diet on cardiometabolic health. Higher adherence to the PDS was associated with lower LDL-C in the Toronto Healthy Diet Study [8], as well as with improvements in cardiometabolic risk factors, including glycated hemoglobin, fasting plasma glucose, triglycerides, waist circumference and BMI, in the PREDIMED-Plus cohort [9].

When looking into the individual components of the Portfolio Diet, plant protein, nuts and viscous fiber was associated with lower mortality risk, whereas plant protein consumption was associated with lower risk of CVD death in our analysis. In the present analysis, the magnitude of association between the Portfolio Diet and all-cause mortality (HR: 0.72, 95% CI = 0.61, 0.86) was comparable to plant protein sources (HR: 0.77, 95% CI = 0.62, 0.95), viscous fiber sources (HR: 0.78, 95% CI = 0.62, 0.98) and nuts (HR: 0.73, 95% CI = 0.58, 0.91). Moreover, plant protein sources had an inverse association with CVD mortality among male (HR: 0.50, 95% CI = 0.30, 0.81) participants. The findings of our present analyses are in line with related studies on the association between plant-based foods, CVD and cancer mortality. For example, evidence from meta-analyses have suggested the consumption of legumes [24], viscous fiber sources [25], and nuts [26] were associated with a lower risk of CVD events. A recent meta-analysis of cohort studies also suggests that higher plant protein intake is associated with a reduced risk of all-cause and CVD-related mortality, while the association was not significant for cancer mortality regardless of sex [27]. However, it is notable that only 2 out of 15 analyses on plant protein and cancer mortality were conducted in China [27]. The geographical variation in diet is likely to be overlooked, and therefore more prospective cohorts should be conducted in Asian regions.

Our subgroup analyses showed that greater adherence to the Portfolio Diet was associated with a lower risk of all-cause and cancer mortality among men only. Similarly, an analysis of the Adventist Health Study 2 found that vegetarian dietary patterns (a combined category of vegan, lacto-ovo–vegetarian, pesco-vegetarian, and semi-vegetarian) were associated with a reduced risk of mortality due to ischemic heart disease or CVD in men but the associations were insignificant in women [28]. Although authors did not investigate the reasons behind this finding, they postulate that the nutrient profile of the dietary patterns might differ by sex [28]. Meanwhile, in our study, the sex-specific adherence to individual Portfolio Diet components might have attributed to the differential associations between PDS and mortality. When compared to female participants, men consumed more food sources of plant protein, nuts, plant sterols and saturated fat/cholesterol, although only plant protein has demonstrated prospective associations with the risk of all-cause and CVD mortality among men. Our findings may suggest that a better adherence to the Portfolio Diet recommendation, e.g., 21.4 g of soy protein per 1000 kcal, can be translated into long-term health benefit [29]. Moreover, men were also more likely to have evidence of stroke and angina at baseline, but the sensitivity analyses suggested that both comorbidities did not have substantial influence on the magnitude of the association. When compared to WHI population [10], the female participants in the present study had a higher consumption of food sources rich in plant protein (9.07 vs. 0.77 servings/day at the highest quintile of intake), which may be attributed to the intake of soy and soy products [30]. Further research is necessary to verify the potential sex-specific associations between PDS and cause-specific mortality and the mechanisms behind them, especially in Asian regions.

### 4.3. Strengths and Limitations

The strengths of our study include the prospective cohort design and over 10 years of follow-up for mortality outcomes. Nevertheless, there are some limitations. First, participants in the OS Study had a higher educational level compared with the general Hong Kong population and, therefore, they may have better dietary habits and improved dietary profile. Second, although the FFQ is considered a common method to collect data of usual dietary intake in large epidemiological studies, the data are self-reported. In addition, the smaller number of CVD mortality events than all-cause and cancer deaths may explain the null association between Portfolio Diet adherence and the risk of CVD mortality. Moreover, diet and lifestyle factors might change over time, and the relevant information was only available at baseline in this current study. Lastly, causation cannot be established because of the observational design, and residual confounding cannot be ruled out.

## 5. Conclusions

Greater adherence to a Portfolio Diet was significantly associated with lower risk of mortality in Hong Kong older adults. These findings have supported the benefits of a Portfolio Diet in reducing the risk of cancer mortality. However, our findings have to be verified in additional populations, including in cohort studies that measured repeated dietary habits over time.

## Figures and Tables

**Table 1 nutrients-13-04360-t001:** Baseline characteristics of participants in Mr. Osteoporosis (OS) and Ms. OS Study.

	Male (*n* = 1996)	Female (*n* = 1995)	*p* Value
	Mean (SD)/*n* (%)		
Age	72.39 ± 5.01	72.59 ± 5.36	*0.23*
Post-secondary Education	286 (14.3%)	130 (6.5%)	*<0.01*
Physical activity (PASE score)	97.37 ± 50.29	85.29 ± 33.12	*<0.01*
Smoking habit			*<0.01*
● Former smoker	1036 (51.9%)	153 (7.7%)	
● Current smoker	237 (11.9%)	37 (1.9%)	
Drink > 12 alcoholic drinks in the past year	471 (23.6%)	51 (2.6%)	*<0.01*
Dietary energy (kcal)	2099.09 ± 586.70	1582.87 ± 461.70	*<0.01*
Plant protein sources (serving)	5.19 ± 4.42	4.03 ± 3.54	*<0.01*
Viscous fiber sources (serving)	3.61 ± 3.41	4.25 ± 2.86	*<0.01*
Nuts (serving)	0.09 ± 0.18	0.06 ± 0.16	*<0.01*
Plant sterols (mg)	361.50 ± 160.51	320.56 ± 168.87	*<0.01*
MUFAs sources (serving)	0.07 ± 0.18	0.08 ± 0.17	*<0.01*
Saturated fat/cholesterol sources (serving)	2.26 ± 1.63	1.27 ± 1.10	*<0.01*
Systolic blood pressure (mmHg)	141.85 ± 19.81	143.41 ± 18.36	*0.01*
Body mass index (kg/m^2^)	23.45 ± 3.13	23.92 ± 3.45	*<0.01*
History of diabetes	293 (14.7%)	286 (14.3%)	*0.76*
History of hypertension	834 (41.8%)	869 (43.6%)	*0.26*
History of stroke	108 (5.4%)	65 (3.3%)	*<0.01*
History of heart attack	200 (10.0%)	192 (9.6%)	*0.67*
History of angina	205 (10.3%)	147 (7.4%)	*<0.01*
History of congestive heart failure	73 (3.7%)	78 (3.9%)	*0.68*
History of cancer	87 (4.4%)	89 (4.5%)	*0.88*

Chi-square (categorical variables) and one-way ANOVA (continuous variables) for subgroup differences. Abbreviations: PASE: Physical Activity Scale for the Elderly; MUFA: Monounsaturated fatty acids; SD: standard deviation.

**Table 2 nutrients-13-04360-t002:** Baseline characteristics of participants in Mr. OS and Ms. OS Study by quartiles of Portfolio Diet scores.

	Mean (SD)/*n* (%)				
	Q1 (<14, *n* = 1168)	Q2 (14–16, *n* = 944)	Q3 (17–19, *n* = 1080)	Q4 (≥20, *n* = 799)	*p* Value
Age	72.76 ± 5.36	72.69 ± 5.08	72.39 ± 5.20	71.99 ± 5.02	*<0.01*
Male	548 (46.9%)	539 (57.1%)	504 (46.7%)	404 (50.6%)	*0.01*
Post-secondary Education	87 (7.4%)	79 (8.4%)	123 (11.4%)	127 (15.9%)	*<0.01*
Physical activity (PASE score)	86.31 ± 39.83	90.22 ± 42.63	94.02 ± 44.84	96.34 ± 44.57	*<0.01*
Smoking habit					*<0.01*
● Former smoker	378 (32.4%)	246 (26.1%)	340 (31.5%)	225 (28.2%)	
● Current smoker	123 (10.5%)	69 (7.3%)	62 (5.7%)	20 (2.5%)	
Drink > 12 alcoholic drinks in the past year	165 (14.1%)	121 (12.8%)	151 (14.0%)	85 (10.6%)	*0.11*
Dietary energy (kcal)	1682.50 ± 518.81	1767.53 ± 574.63	1932.87 ± 587.90	2035.57 ± 618.66	*<0.01*
Plant protein sources (serving)	2.83 ± 2.08	4.19 ± 3.22	5.23 ± 3.62	6.90 ± 5.92	*<0.01*
Viscous fiber sources (serving)	2.29 ± 1.85	3.55 ± 3.23	4.52 ± 2.75	5.98 ± 3.68	*<0.01*
Nuts (serving)	0.03 ± 0.07	0.06 ± 0.10	0.09 ± 0.14	0.16 ± 0.30	*<0.01*
Plant sterols (mg)	231.05 ± 95.01	314.37 ± 135.89	390.92 ± 158.44	465.88 ± 176.85	*<0.01*
MUFAs sources (serving)	0.01 ± 0.03	0.03 ± 0.10	0.08 ± 0.16	0.24 ± 0.26	*<0.01*
Saturated fat/cholesterol sources (serving)	1.92 ± 1.49	1.70 ± 1.50	1.60 ± 1.37	1.77 ± 1.47	*<0.01*
Systolic blood pressure (mmHg)	142.68 ± 18.78	142.82 ± 19.08	142.91 ± 19.82	141.97 ± 19.11	*0.73*
Body mass index (kg/m^2^)	23.60 ± 3.50	23.89 ± 3.31	23.62 ± 3.17	23.67 ± 3.14	*0.18*
History of diabetes	162 (13.9%)	145 (15.4%)	162 (15.0%)	110 (13.8%)	*<0.01*
History of hypertension	504 (43.2%)	408 (43.2%)	447 (41.4%)	344 (43.1%)	*0.80*
History of stroke	46 (3.9%)	46 (4.9%)	48 (4.4%)	33 (4.1%)	*0.75*
History of heart attack	123 (10.5%)	84 (8.9%)	91 (8.4%)	94 (11.8%)	*0.06*
History of angina	110 (9.4%)	83 (8.8%)	86 (8.0%)	73 (9.1%)	*0.66*
History of congestive heart failure	42 (3.6%)	40 (4.2%)	44 (4.1%)	25 (3.1%)	*0.61*
History of cancer	47 (4.0%)	46 (4.9%)	45 (4.2%)	38 (4.8%)	*0.74*

Chi-square (categorical variables) and one-way ANOVA (continuous variables) for subgroup differences. Abbreviations: PASE (Physical Activity Scale for the Elderly); Q (Quartile); SD: standard deviation.

**Table 3 nutrients-13-04360-t003:** Prospective association of the Portfolio Diet Score with the risk of mortality outcomes among 3991 participants in Mr. OS and Ms. OS Study.

			All-Cause Mortality		CVD Mortality		Cancer Mortality
	Person-Years	No. of Deaths	HR (95% CI)	No. of Deaths	HR (95% CI)	No. of Deaths	HR (95% CI)
**Overall**							
Q1 (*n* = 1168)	13,895	458 (39.2%)	1.00	100 (8.6%)	1.00	162 (13.9%)	1.00
Q2 (*n* = 944)	11,772	311 (32.9%)	0.87 (0.75, 1.01)	71 (7.5%)	0.91 (0.67, 1.24)	107 (11.3%)	0.84 (0.66, 1.08)
Q3 (*n* = 1080)	13,424	388 (35.9%)	0.96 (0.84, 1.10)	84 (7.8%)	0.98 (0.73, 1.31)	128 (11.9%)	0.90 (0.71, 1.14)
Q4 (*n* = 799)	10,317	213 (26.7%)	0.72 (0.61, 0.86) *	59 (7.4%)	0.90 (0.64, 1.26)	72 (9.0%)	0.72 (0.54, 0.96) *
*p*-value for trend			*<0.01*		*0.64*		*0.05*
*p*-value for non-linearity			*0.16*		*0.30*		*0.64*
**Male**							
Q1 (*n* = 620)	7119	300 (48.4%)	1.00	58 (9.4%)	1.00	103 (16.6%)	1.00
Q2 (*n* = 405)	4892	173 (42.7%)	0.94 (0.78, 1.14)	38 (9.4%)	1.10 (0.73, 1.66)	68 (16.8%)	1.04 (0.76, 1.42)
Q3 (*n* = 576)	6936	259 (45.0%)	1.02 (0.86, 1.20)	56 (9.7%)	1.14 (0.79, 1.66)	93 (16.1%)	1.09 (0.82, 1.46)
Q4 (*n* = 395)	5113	120 (30.4%)	0.63 (0.51, 0.79) *	36 (9.1%)	0.90 (0.58, 1.39)	35 (8.9%)	0.59 (0.39, 0.87) *
*p*-value for trend			*<0.01*		*0.85*		*0.08*
*p*-value for non-linearity			*0.01*		*0.07*		*0.05*
**Female**							
Q1 (*n* = 548)	6776	158 (28.8%)	1.00	42 (7.7%)	1.00	59 (10.8%)	1.00
Q2 (*n* = 539)	6880	138 (25.6%)	0.81 (0.64, 1.02)	33 (6.1%)	0.75 (0.47, 1.18)	39 (7.2%)	0.65 (0.43, 0.98) *
Q3 (*n* = 504)	6488	129 (25.6%)	0.84 (0.66, 1.08)	28 (5.6%)	0.78 (0.48, 1.28)	35 (6.9%)	0.62 (0.40, 0.95) *
Q4 (*n* = 404)	5204	93 (23.0%)	0.88 (0.67, 1.15)	23 (5.7%)	0.92 (0.54, 1.56)	37 (9.2%)	0.87 (0.57, 1.34)
*p*-value for trend			*0.34*		*0.65*		*0.35*
*p*-value for non-linearity			*0.25*		*0.54*		*<0.01*

Abbreviations: CVD, cardiovascular disease; HR, hazard ratio. * *p* < 0.05; The Cox regression model was adjusted for sex, age, dietary energy, body mass index, physical activity, systolic blood pressure, medical history (diabetes, hypertension, stroke, heart attack, angina, congestive heart failure or cancer), smoking habit, alcohol drinking, education level.

**Table 4 nutrients-13-04360-t004:** Associations between the highest to lowest adherence to the individual components of the Portfolio Diet and risk of mortality.

	All-Cause Mortality	CVD Mortality	Cancer Mortality
	HR (95% CI)	HR (95% CI)	HR (95% CI)
**Male**			
Plant protein sources	0.77 (0.62, 0.95) *	0.50 (0.30, 0.81) *	0.98 (0.69, 1.39)
Viscous fiber sources	0.78 (0.62, 0.98) *	1.16 (0.73, 1.85)	0.71 (0.48, 1.04)
Nuts	0.73 (0.58, 0.91) *	0.91 (0.57, 1.46)	0.77 (0.54, 1.12)
Plant sterols	0.87 (0.68, 1.10)	0.88 (0.53, 1.47)	0.91 (0.61, 1.34)
MUFAs sources	0.94 (0.78, 1.13)	1.14 (0.79, 1.65)	0.84 (0.60, 1.18)
Saturated fat/cholesterol sources	0.93 (0.74, 1.18)	1.06 (0.64, 1.75)	0.85 (0.56, 1.29)
**Female**			
Plant protein sources	0.90 (0.68, 1.19)	1.03 (0.57, 1.84)	0.70 (0.43, 1.13)
Viscous fiber sources	0.81 (0.61, 1.07)	1.17 (0.67, 2.05)	0.81 (0.50, 1.32)
Nuts	0.97 (0.72, 1.30)	0.63 (0.33, 1.19)	0.97 (0.59, 1.61)
Plant sterols	0.99 (0.72, 1.36)	1.36 (0.72, 2.58)	0.76 (0.44, 1.29)
MUFAs sources	1.12 (0.91, 1.40)	0.83 (0.51, 1.34)	1.18 (0.82, 1.70)
Saturated fat/cholesterol sources	1.18 (0.86, 1.61)	1.53 (0.76, 3.08)	1.08 (0.63, 1.84)

Abbreviations: CVD, cardiovascular disease; HR, hazard ratio. * *p* < 0.05; The Cox regression model was adjusted for sex, age, dietary energy, body mass index, physical activity, systolic blood pressure, medical history (diabetes, hypertension, stroke, heart attack, angina, congestive heart failure or cancer), smoking habit, alcohol drinking, education level.

## Data Availability

The data presented in this study are available on request from the corresponding author.

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
