# Peer review of "Prospective Association of the Portfolio Diet with All-Cause and Cause-Specific Mortality Risk in the Mr. OS and Ms. OS Study"

_nutrients, 2021, doi:10.3390/nu13124360_

Round 1
Reviewer 1 Report
Prospective Association of the Portfolio Diet with All-cause 2 and Cause-specific Mortality Risk in the Mr. OS & Ms. OS 3 Study
Kenneth Lo , et al aimed to “evaluated how the adherence to the Portfolio Diet may have sex-specific associations with the risk of all-cause, CVD and cancer mortality among 63 Hong Kong older adults that participated in the Mr. OS & Ms. OS study.”
The paper is well written but nevertheless there are some remarks to make:
pag 3 2.5. Statistical Analysis:
(righe 128-129): Cox proportional hazard regression models were used to identify the association between the PDS quartiles and mortality outcome. But no mention is made of the actual proportional hazard that is a necessary condition for the applicability of the Cox model.
It would be appropriate, even as supplementary material, to include a graph or statistical test showing the proportionality of the risks.
When studying causes specific of death (CVD and Cancer for example), the Cox model is not indicated, risk estimates are not accurate, but a Competitive Risk Model is more appropriate.
After fitting survival model, it would be useful to plot the estimated hazard.
When estimating the risk of death in subjects it is more useful to use the longitudinal age (age at death) as the survival time, rather than the observation period from enrolment to the end of the study or the subject's death. Each subject carries with him or her an exposure to death, or the various causes of death, that begins before the period of enrolment.
Table 1 e table 2:
If the exact pvalue is given, it is not necessary to put an asterisk to indicate that the pvalue is <0.05.
p value should be written in italics (pvalue).
Table 3 e table 4:
p value should be written in italics (pvalue).Having reported the value of the 95% C.I, I would advise not to report the pvalue, but the significance could be indicated with asterisks.
For example :
Plant protein sources 0.77*( 0.62; 0.95) instead of reporting pvalue 0.02*. If the exact pvalue is given, it is not necessary to put an asterisk to indicate that the pvalue is <0.05.
Author Response
Kenneth Lo et al aimed to evaluate how the adherence to the Portfolio Diet may have sex-specific associations with the risk of all-cause, CVD and cancer mortality among Hong Kong older adults that participated in the Mr. OS & Ms. OS study. The paper is well written but nevertheless there are some remarks to make.
Page 3 2.5. Statistical Analysis:
(line 128-129): Cox proportional hazard regression models were used to identify the association between the PDS quartiles and mortality outcome. But no mention is made of the actual proportional hazard that is a necessary condition for the applicability of the Cox model. It would be appropriate, even as supplementary material, to include a graph or statistical test showing the proportionality of the risks.
Response: Thank you for the suggestion. We have tested the proportionality of the risks according to outcomes (all-cause, CVD or cancer mortality) and sex (male, female) and have presented the results in Supplementary Figure 2. As revealed by Schoenfeld test in Supplementary Figure 2, most Cox regression models did not violate the assumption except for the association between PDS and all-cause mortality among all participants (p=0.0199).
When studying causes specific of death (CVD and Cancer for example), the Cox model is not indicated, risk estimates are not accurate, but a Competitive Risk Model is more appropriate.
Response: Thank you for the suggestion. As an additional analysis (Supplementary Table 3), we have used the competitive risk model to account for the competing nature of CVD and cancer mortality, then compared the magnitude of association with that obtained from the standard Cox models (Table 3). Since the magnitude of association as presented in both tables was virtually the same, we kept Cox model in the main analyses.
After fitting survival model, it would be useful to plot the estimated hazard.
Response: We have plotted the estimated hazard as Supplementary Figure 1, and did not find significant differences in incidence by sex.
When estimating the risk of death in subjects it is more useful to use the longitudinal age (age at death) as the survival time, rather than the observation period from enrolment to the end of the study or the subject's death. Each subject carries with him or her an exposure to death, or the various causes of death, that begins before the period of enrolment.
Response: Thank you for the comment. While participants may adhere to the Portfolio Diet in a certain extent before the period of enrolment, the level of adherence is uncertain. Therefore, we propose to use the Portfolio Diet adherence upon enrolment to associate with the mortality risk from enrolment to the end of the study. As a sensitivity analysis (Supplementary Table 2), we have used the longitudinal age as the survival time, and the magnitude of association was virtually the same.
Table 1 and table 2:
If the exact p-value is given, it is not necessary to put an asterisk to indicate that the p-value is <0.05. p value should be written in italics (p-value).
Response: We have revised as suggested.
Table 3 and table 4:
p value should be written in italics (p-value). Having reported the value of the 95% C.I, I would advise not to report the p-value, but the significance could be indicated with asterisks.
For example: Plant protein sources 0.77*(0.62; 0.95) instead of reporting p-value 0.02*. If the exact p-value is given, it is not necessary to put an asterisk to indicate that the p-value is <0.05.
Response: We have revised as suggested. We use the format “0.77 (0.62; 0.95) *” so that the asterisks can line up better across the columns.
Reviewer 2 Report
Abstract
Line 22 – Spell out CVD;
Line 27 and 30 - The conclusion “These findings supported the benefits of the Portfolio Diet in reducing the risk of mortality, particularly from cancer” is not coherent with the result “most associations between the Portfolio Diet and mortality were not significant among females”; Please rewrite it: I suggest using your information in line 252 “while the protection 252 for cancer mortality risk might reach the plateau at the highest quartile of PDS for females”, at least for cancer.
Introduction
Line 45 - It should be indicated if the Portfolio Diet contains or not any foods of animal origin;
Lines 61 and 64 – Spell out WHI and please indicate better what is the Mr. OS & Ms. OS study;
Line 63 – Please justify the choice of the study population. Is there a great incidence of the mentioned diseases in Hong Kong adults? Are there differences between sexes in this incidence? In line 70 it is mentioned that the study was performed in persons aged at least 65 years; hence, the previous reference simply as “adults” is not quite correct;
Line 78 – Spell out FFQ;
Lines 87 to 101 – Indicate in case this methodology is previously validated, or if there are other studies in the literature applying it;
Line 123- Smoking habit, alcohol consumption, physical activity, dietary intake and self-reported history of chronic diseases are not demographic information;
Results
Table 1 – The items should follow an order, with dietary characteristics together and health items together; the same for table 2;
Line 206 – the method to evaluate all-cause mortality should be better explained; what is the difference between all cause mortality and total mortality?;
Discussion
Line 262 – the fact that the WHI study was performed in postmenopausal women should be also consider n the interpretation of these results;
Line 292 - Vegetarian diet here is vegetarian including eggs and milk, or vegan?
Author Response
Line 22 – Spell out CVD;
Response: We have revised the term as “cardiovascular disease”.
Line 27 and 30 - The conclusion “These findings supported the benefits of the Portfolio Diet in reducing the risk of mortality, particularly from cancer” is not coherent with the result “most associations between the Portfolio Diet and mortality were not significant among females”; Please rewrite it. I suggest using your information in line 252 “while the protection for cancer mortality risk might reach the plateau at the highest quartile of PDS for females”, at least for cancer.
Response: Thank you for the suggestion, we have added a statement “The protection for cancer mortality risk might reach the plateau at the highest adherence to the Portfolio Diet for females”.
Introduction
Line 45 - It should be indicated if the Portfolio Diet contains or not any foods of animal origin
Response: As now being indicated in introduction, Portfolio Diet contains a list of plant-based foods or nutrients with cholesterol-lowering properties.
Lines 61 and 64 – Spell out WHI and please indicate better what is the Mr. OS & Ms. OS study
Response: WHI is the abbreviation of Women’s Health Initiative. We have added this abbreviation at its first appearance in the previous paragraph. The full name of Mr. OS & Ms. OS study is Mr Osteoporosis and Ms Osteoporosis Study, which is explained at the beginning of method section.
Line 63 – Please justify the choice of the study population. Is there a great incidence of the mentioned diseases in Hong Kong adults? Are there differences between sexes in this incidence?
Response: Thank you for the question. There are two reasons for our choice. First, the health benefit of Portfolio Diet has not been verified in Chinese population. Second, Hong Kong is an ageing Chinese population with high life expectancy. In an analysis of the secular trends of cancer incidences in 1983–2017, the total incident cancer case counts increased 156.5% for women and 96% for men, which might due to population growth (66.1% for women, 25.4% for men) and population ageing (95% for women, 119.4% for men). That would be important to evaluate how a healthy dietary pattern may reduce the mortality risk due to chronic diseases for older adults.
Reference: Cancers 2021, 13(22), 5727; https://doi.org/10.3390/cancers13225727
In line 70 it is mentioned that the study was performed in persons aged at least 65 years; hence, the previous reference simply as “adults” is not quite correct.
Response: Thank you for the comment. We have used the term “older adults” throughout the manuscript, which refers to the elderly that participated in our study.
Line 78 – Spell out FFQ
Response: Thank you for the reminder. We have provided the abbreviation for FFQ (food frequency questionnaire) at its first appearance in the third paragraph in introduction, “A Portfolio Diet score (PDS) has been developed and validated to assess the adherence to the Portfolio Diet using food frequency questionnaires (FFQ)…”.
Lines 87 to 101 – Indicate in case this methodology is previously validated, or if there are other studies in the literature applying it
Response: As now being added into the same paragraph, the adherence score of Portfolio Diet has been validated in 652 participants in the Toronto Healthy Diet Study, which has shown inverse association with lower LDL-C, and showed reasonable agreement between the score from the FFQ and 7-day dietary record.
Line 123- Smoking habit, alcohol consumption, physical activity, dietary intake and self-reported history of chronic diseases are not demographic information.
Response: Thank you the for comment, we have revised the term as “demographic, lifestyle and health information”.
Results
Table 1 – The items should follow an order, with dietary characteristics together and health items together; the same for table 2
Response: Thank you for the suggestion. In both tables, we start with demographics (age, sex, education), followed by lifestyle factors (physical activity, smoking and alcohol drinking), dietary characteristics, and lastly the health profile (blood pressure, BMI and medical history).
Line 206 – the method to evaluate all-cause mortality should be better explained; what is the difference between all-cause mortality and total mortality?
Response: Thank you for the reminder, as stated in the first paragraph of statistical analysis, “to examine the potential non-linear relationship between the PDS and the risk of mortality, we compared the fit of continuous models with or without cubic spline terms of PDS. A likelihood test with p < 0.05 would suggest a better fit regression model by including the quadratic term, hence a non-linear relationship between the PDS and all-cause, CVD or cancer mortality”. Besides, we now use the term “all-cause mortality” throughout the manuscript for consistency.
Discussion
Line 262 – the fact that the WHI study was performed in postmenopausal women should be also consider in the interpretation of these results.
Response: Thank you for the reminder, we have revised the phrase as “3,991 older adults when compared with 123,330 postmenopausal women in WHI” so readers can understand the context of previous study.
Line 292 - Vegetarian diet here is vegetarian including eggs and milk, or vegan?
Response: According to the paper, it was a combined category of vegan, lacto-ovo–vegetarian, pesco-vegetarian, and semi-vegetarian. We have explained it in the revised manuscript.
Round 2
Reviewer 1 Report
The corrections I suggested have been made. In Supplementary Table 3, since an analysis of competitive risks has been made, they are no longer HR (Hazard Ratio) but SHR (Subdistribution Hazard Ratio).
Author Response
The corrections I suggested have been made. In Supplementary Table 3, since an analysis of competitive risks has been made, they are no longer HR (Hazard Ratio) but SHR (Subdistribution Hazard Ratio).
Response: Thank you for the suggestion. We have revised the abbreviation and the footnote in Supplementary Table 3. In the main text, we have added the abbreviation of SHR at its first appearance in the last paragraph of statistical analysis, and in 3.3 Sensitivity analysis and competing risk analysis.